# The Ethyl Acetate Extract of *Caulerpa microphysa* Promotes Collagen Homeostasis and Inhibits Inflammation in the Skin

**Kuo-Yun Lu** [1,†], **Li-Ching Cheng** [1,2,†], **Zheng-Ci Hung** [3], **Ze-Ying Chen** [4], **Chuang-Wei Wang** [5] and **Hsin-Han Hou** [3,6,7,*]

1   Department of Nursing, Division of Basic Medical Sciences, Chang-Gung University of Science and Technology, Taoyuan 333, Taiwan; kylu.tmb@gmail.com (K.-Y.L.); victoria@gw.cgust.edu.tw (L.-C.C.)
2   Department of General Surgery, Chang Gung Memorial Hospital at Linkou, Taoyuan 333, Taiwan
3   Graduate Institute of Oral Biology, School of Dentistry, National Taiwan University, Taipei 100, Taiwan; nayuta_koto@hotmail.com
4   Institute of Environmental and Occupational Health Sciences, College of Public Health, National Taiwan University, Taipei 100, Taiwan; zeyingchen@ntu.edu.tw
5   Department of Dermatology, Drug Hypersensitivity Clinical and Research Center, Chang Gung Memorial Hospital, Taoyuan 333, Taiwan; kiruamairo@gmail.com
6   Graduate Institute of Clinical Dentistry, School of Dentistry, National Taiwan University, Taipei 100, Taiwan
7   Department of Dentistry, National Taiwan University Hospital, Taipei 100, Taiwan
*   Correspondence: houhh@ntu.edu.tw
†   These authors contributed equally to this work.

**Abstract:** Inflammation and collagen-degrading enzymes' overexpression promote collagen decomposition, which affects the structural integrity of the extracellular matrix. The polysaccharide and peptide extracts of the green alga *Caulerpa microphysa* (*C. microphysa*) have been proven to have anti-inflammatory, wound healing, and antioxidant effects in vivo and in vitro. However, the biological properties of the non-water-soluble components of *C. microphysa* are still unknown. In the present study, we demonstrated the higher effective anti-inflammatory functions of *C. microphysa* ethyl acetate (EA) extract than water extract up to 16–30% in LPS-induced HaCaT cells, including reducing the production of interleukin (IL)-1β, IL-6, IL-8, and tumor necrosis factor-α (TNF-α). Furthermore, the excellent collagen homeostasis effects from *C. microphysa* were proven by suppressing the matrix metalloproteinase-1 (MMP-1) secretion, enhancing type 1 procollagen and collagen expressions dose-dependently in WS1 cells. Moreover, using UHPLC-QTOF-MS analysis, four terpenoids, siphonaxanthin, caulerpenyne, caulerpal A, and caulerpal B, were identified and may be involved in the superior collagen homeostasis and anti-inflammatory effects of the *C. microphysa* EA extract.

**Keywords:** *Caulerpa microphysa*; ethyl acetate extract; lipopolysaccharide; inflammation; collagen; matrix metalloproteinase-1; terpenoids

## 1. Introduction

Seaweed, also known as marine macroalgae, mainly grows in coastal intertidal zones and most types of seaweed are common edible sea vegetables for coastal residents. The intertidal habitat is harsh, especially in the high- and mid-tidal areas. Green algae, which comprise the main seaweed growing in the high- and mid-tidal zones must adapt to daily tidal changes and resist strong sunlight and high temperature. The green algae produce many adaptive bioactive compounds and metabolites to protect against environmental damage [1–3], including phenolic compounds [4,5], pigments [6,7], polysaccharides [8–10], terpenoids [11–13], etc.

In recent years, research has shown that these active metabolites of algae also exhibit the potential for a variety of pharmacological applications. For example, water-soluble polysaccharides from some green algae have been discovered to have antioxidant and

immunomodulatory activities [14,15]. A variety of water-insoluble terpenoids from seaweed also have been identified to have diverse physiological activities, such as carotenoids, tetraterpene pigments, which are the most widely distributed pigments in nature and are also found in large amounts in photosynthetic seaweeds. Their conjugated double-bond structures are considered extremely effective natural antioxidants because of their ability to scavenge reactive oxygen species [16,17]. Siphonaxanthin is a marine carotenoid that exists in green algae. It is an oxidative metabolite of lutein. Its structure is similar to lutein but with one keto group located at C-8 and an extra hydroxyl group at C-19 [17]. Siphonaxanthin has been shown to have angiogenesis-inhibiting, anti-inflammatory, and lipid metabolism-regulating activities [12,18–21]. Caulerpenyne, a sesquiterpene metabolite of *Caulerpa*, has shown antioxidant, antiproliferative, antibiotic, and antiviral activities [13,22–24]. Caulerpal A and B are two sesquiterpenes isolated from *Caulerpa taxifolia* with an unusual aromatic pentene carbon skeleton and caulerpin. These two metabolites exhibit a broad spectrum of antifungal activity [25].

The green alga *Caulerpa microphysa* (*C. microphysa*) is an edible green alga native to the intertidal zones of Taiwan, China, and the Philippines. *C. microphysa* belongs to the green algae genus *Caulerpa*. The thallus is composed of a horizontal stolon connected by rhizoids and erect stems connected by rounded vesicular fronds. Its appearance is similar to *C. lentillifera* but with shorter erect stems and smaller bulbous fronds. *C. microphysa* grows all year round but grows slowly in summer due to the hot sun and high temperatures. Previous studies have found that its polysaccharide extract has anti-inflammatory and wound healing effects in 3T3-L1 fibroblasts [26], as well as antioxidant and anticoagulant effects in an ex vivo study [27]. Peptides extracted from *C. microphysa* also exhibit antitumor properties in human promyelocytic leukemia cells (HL-60) [28]. However, there is a lack of literature mentioning the effect of *C. microphysa* on dermatology and discussing the physiological activities of the non-water-soluble components of this edible alga.

Type I collagen is one of the most abundant molecules in the body, especially in the skin, bones, and connective tissues [29]. In the skin, it accounts for three-quarters of the dry weight of skin and is highly expressed in fibroblasts [30,31]. When fibroblasts complete triple-helical procollagen synthesis, the procollagen is secreted into the extracellular matrix (ECM) and cleaves the carboxy terminals and the amino terminals through proteinases. The terminal-cutting procollagen chains, tropocollagen, combine with each other to form collagen fibers [32–34]. Collagen homeostasis means the stability of collagen content in ECM, which is determined by the balance of synthesis and degradation. Previous studies have indicated that collagen is degraded during various normal physiological processes, such as wound healing and skin aging. In addition, pathological conditions such as inflammation, atherosclerotic cardiovascular disease, and sunburn accelerate collagen degradation [35–37]. Members of the MMP family are involved in collagen degradation [38,39], especially MMP-1, known as interstitial collagenase, which is primarily responsible for the degradation of dermal type I collagen. MMP-1 is expressed in unstimulated fibroblasts and upregulated by inflammation and photoaging [40–42].

The aim of this study was to clarify whether the non-water-soluble components of *C. microphysa* have physiological effects on the inflammation and collagen homeostasis of skin cells and to further analyze which potential secondary metabolites are present in the water-insoluble extract of *C. microphysa*. In a comparison of the anti-inflammatory effects between the water extract and EA extract of *C. microphysa* on keratinocytes, surprisingly, the EA extract is more effective than the water extract in inhibiting LPS-induced IL-1β, IL-6, IL-8, and TNF-α expressions. For human fibroblasts, the EA extract of *C. microphysa* also exhibits superior collagen homeostasis effects by increasing type I procollagen and collagen expressions and suppressing MMP-1, the interstitial collagen-degrading enzyme, secretion. By UHPLC-QTOF-MS analysis, we observed the presence of four terpenoid compounds, which are siphonaxanthin, caulerpenyne, caulerpal A, and caulerpal B in the *C. microphysa* EA extract. Whether these four terpenoid compounds are involved in the anti-inflammatory and collagen-homeostasis effects of *C. microphysa* EA extract remains to be

further clarified. This experiment was the first to verify the superior anti-inflammatory and collagen-homeostasis effects of the water-insoluble extract of *C. microphysa* on dermatology and to analyze its potential secondary metabolites.

## 2. Materials and Methods

### 2.1. Cells

A human epidermal keratinocyte cell line, HaCaT (300493, Cell Lines Service GmbH, Eppelheim, Deutschland), was cultured in DMEM (with 10% FBS, 2 mM L-glutamine, 4.5 g/L glucose, and 1% Penicillin–Streptomycin–Neomycin Solution) in a 37 °C oven with 5% $CO_2$. Human skin fibroblast, WS1 cells (60300, BCRC), was cultured in MEM (with 10% FBS and 1% Penicillin–Streptomycin–Neomycin Solution) in a 37 °C oven with 5% $CO_2$. The split ratio in both cells was 1:3. Finally, 0–200 ng/mL LPS was applied to trigger cell inflammation.

### 2.2. Preparation of Caulerpa microphysa Extracts

The *Caulerpa microphysa* (*C. microphysa*), after morphological identification, was cultivated in Penghu waters, Taiwan, and harvested in autumn. After washing with reverse osmosis water, it was homogenized 3 times at 12,000 rpm for 5 min. The homogenate was filtrated and centrifuged with $6000 \times g$ at 4 °C for 10 min; then, we collected the supernatant (the supernatant/fresh specimens was about 55%). The supernatant was filtered using a 0.22 μm filter; then, we added EA to the filtrate at a ratio of 1:1 by weight and used a separatory funnel. After violently shaking to mix, we let it stand to separate the lower water layer (WL) and upper EA layer (EAL). Each layer was collected using five repeats for separation. Then, the EAL was concentrated with a reduced pressure vacuum concentrator (BUCHI/ROTAVAPOR R-300, Taipei, Taiwan) in 80 mbar and 45 °C conditions. The yield was 0.015%. The WL was dried using a freeze dryer (Yihua/FDS-5B-0.5-LT, New Taipei City, Taiwan).

### 2.3. Ultra-Performance Liquid Chromatography (UPLC) and Quadrupole Time-of-Flight Mass Spectrometry (QTOF)

We took the 17-fold concentrated EAL of *C. microphysa* for UHPLC-QTOF MS analysis. The analysis was performed using an Agilent 1290 Infinity II LC System (Agilent Technologies, Santa Clara, CA, USA) coupled with an Agilent 6545 LC/QTOF MS. The extracts were separated on a Waters BEH C18 column (100 × 2.1 mm, 1.7 μm). The mobile phases consisted of (A) 5 mM ammonium acetate$_{(aq)}$ and (B) methanol. The liquid chromatography gradient started at 5% of (B) for 30 s and increased to 60% of (B) at 7 min; then, it turned to 100% of (B) at 14 min, which was held for 3 min; then, it was decreased to the initial proportion and re-equilibrated for 2.5 min. The flow rate was 0.3 mL/min, the injection volume was 6 μL, and the column temperature was 40 °C. The QTOF MS was conducted on an Agilent Jet Stream ESI positive mode, with a mass range of *m/z* 100 to 1500 at a scan rate of 3 spectra per second. Accurate mass measurements by the instrument were ensured using an automated calibrant delivery system that continuously introduced a reference solution with a mass mix of *m/z* 121.050873 (purine) and *m/z* 922.009798 (HP-0921) in the ESI-positive mode. The parameters set for ESI-MS included the following: drying gas temperature: 325 °C; drying gas flow rate: 9 L/min; nebulizer gas pressure: 45 psi; sheath gas temperature: 350 °C; sheath gas flow: 11 L/min; capillary voltage: 4500 V; fragmentor voltage: 175 V; radiofrequency voltage in the octupole: 750 V; and fixed collision energies of 0 eV, 20 eV, and 40 eV. Data acquisition was performed on Mass Hunter Workstation Software Data Acquisition for Q-TOF, version B.06.01.

### 2.4. WST-1

The cellular number was evaluated by WST-1 assay (Roche, Basel, Switzerland) based on the manufacturer's instruction. At specified time intervals, the cultured cells underwent a pulsing regimen with the oxidation–reduction indicator WST-1 (10% *vol/vol*) for a duration

of 4 h. The color development (A450 nm–A600 nm) was then quantified to assess the activity of mitochondrial dehydrogenases. This activity was deemed proportional to the quantity of viable cultured cells and expressed as the cell proliferating index [43]. The outcome following pre-incubation (on 0 h) served to adjust the results as a ratio.

### 2.5. Enzyme-Linked Immunosorbent Assay (ELISA)

Cellular media from the cells were analyzed by IL-1b (DLB50), IL-6 (D6050B), IL-8 (D8000C), TNF-$\alpha$ (DTA00D), MMP-1 (DY901B), and type-1 collagen (DY6220-05) kits (R & D system, Minneapolis, MN, USA) according to the manufacturer's instructions. The limits of detection of the above kits were 3.9, 3.1, 31.2, 15.6, 62.5, and 31.2 pg/mL, respectively.

### 2.6. Western Blot

Cells were lysed using RIPA lysis buffer (Genestar, Taipei, Taiwan) containing 1% NP-40, 0.1% SDS, 150 mM sodium chloride, 0.5% sodium deoxycholate, and 50 mM Tris. The cell lysates were centrifuged at 12,000 rpm for 5 min before the supernatant was collected. The extracted protein was quantified by protein assay. Equal amounts of protein were separated using 10% SDS-polyacrylamide gel electrophoresis and transferred to Immobilon-P membranes (Millipore, Burlington, MA, USA). After blocking with 5% skimmed milk, the membranes were incubated with Collagen I pAb (A16891) (ABclonal, Woburn, MA, USA) (1:1000) and then incubated with the Goat anti-Rabbit IgG (H + L)-HRP (C04003, Croyez, Taipei, Taiwan) (1:5000). The protein bands were detected using an Immobilon Western Chemiluminescent HRP Substrate (Millipore, Burlington, MA, USA) and quantified by the ImageQuant 5.2 software (GE Healthcare Bio-Sciences, Pittsburgh, PA, USA).

### 2.7. Statistical Analyses

We used the two-tailed Student's *t*-test to compare the difference between two groups with different treatments. For the comparisons among data from more than two experimental conditions, we first used ANOVA to check the existence of differences; then, post hoc Tukey's test was used to identify significant differences between two specific groups. A *p* value less than 0.05 was considered statistically significant.

## 3. Results

The cellular viability of HaCaT was tested under various doses of LPS, EAL, or WL extracts of *C. microphysa*. The results of WST-1 indicated that 200 ng/mL LPS and a high dose of *C. microphysa*'s EAL and WL (200 mg/mL) affected HaCaT cell viability (Figure 1). Therefore, we chose 100 mg/mL as the maximum treating concentration for both *C. microphysa* extracts and 100 ng/mL for the LPS treatment, which had triggered HaCaT cells to secrete inflammatory markers such as IL-1 β, IL-6, IL-8, and TNF-$\alpha$ (Figure 2).

To compare the anti-inflammatory effects between the EAL and WL extracts of *C. microphysa* in LPS-induced keratinocyte inflammation, we pretreated HaCaT cells with 100 mg/mL of EAL or WL extraction of *C. microphysa* for 2 h; then, we treated with 100 ng/mL LPS (L). The results of the ELISA assay showed that both the EAL and WL extracts of *C. microphysa* could attenuate LPS-increased IL-1 β, IL-6, IL-8, and TNF-$\alpha$ secretion (Figure 3). Notably, the *C. microphysa* EA extract significantly inhibited LPS-induced IL-1 β, IL-6, IL-8, and TNF-$\alpha$ more than the water extract by up to 16–30% in HaCaT cells. This result implies that the anti-inflammatory components of *C. microphysa* should be abundant in the water-insoluble layer.

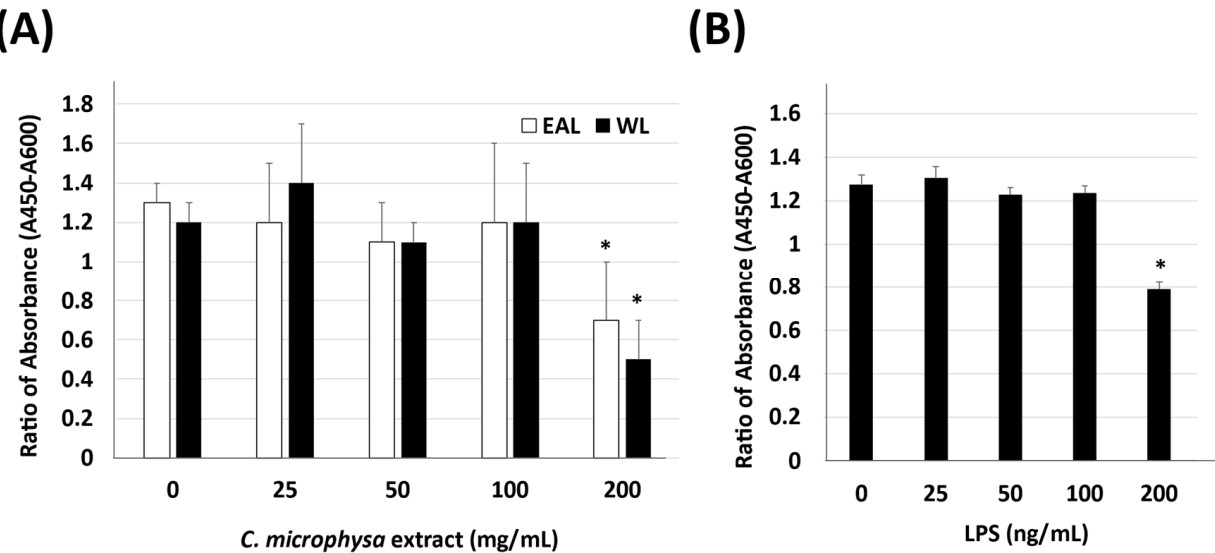

**Figure 1.** Cytotoxicity assay for *C. microphysa's* ethyl acetate layer (EAL) and water layer (WL) extraction and lipopolysaccharide (LPS) treatments. After pretreating HaCaT cells with various doses of the (**A**) EAL, WL, and (**B**) LPS for 24 h, the cellular lysates were collected for the WST-1 assay. (* *p* < 0.05 vs. corresponding zero control).

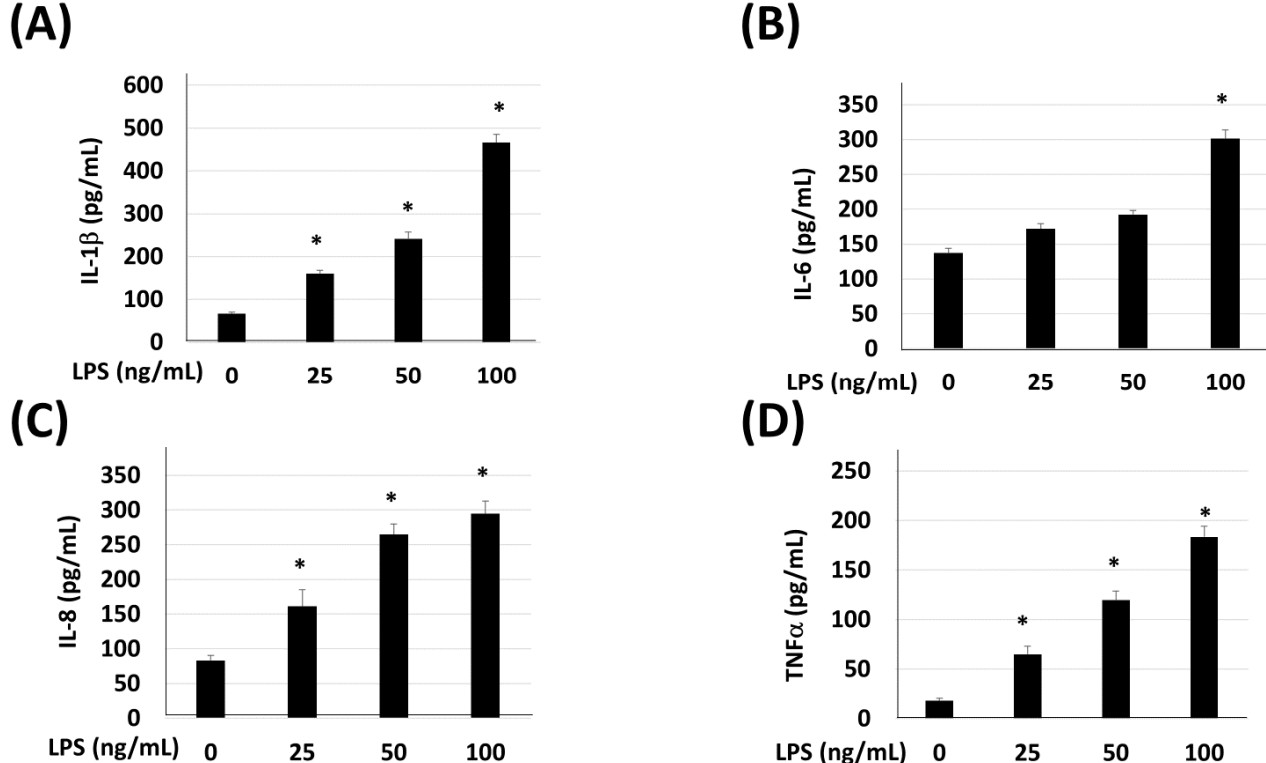

**Figure 2.** LPS triggers IL-1β, IL-6, IL-8, and TNF-α secretion. HaCaT cells were treated with various doses of LPS (ng/mL) for 24 h. The collected cellular medium was applied to (**A**) IL-1β, (**B**) IL-6, (**C**) IL-8, and (**D**) TNF-α ELISA assays. (* *p* < 0.05 vs. corresponding zero control).

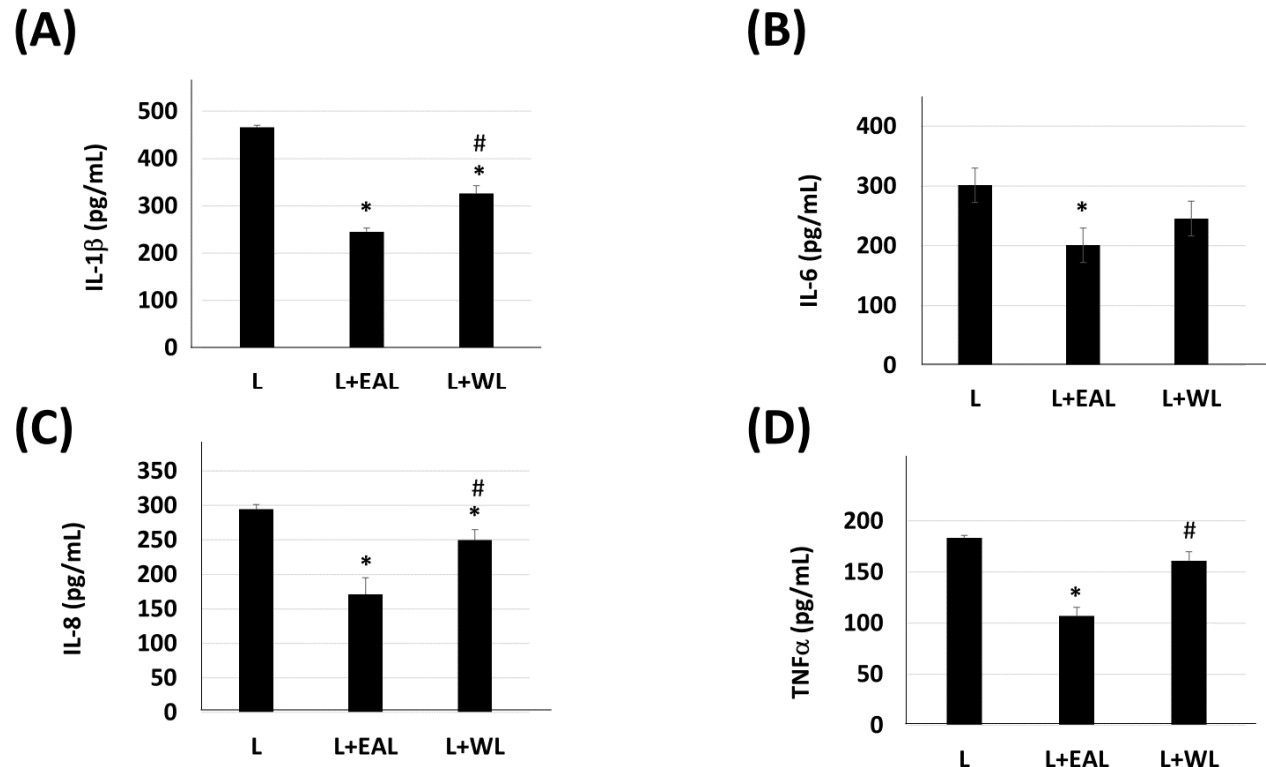

**Figure 3.** EAL extract of *C. microphysa* inhibits LPS-induced IL-1β, IL-6, IL-8, and TNF-α secretion more significantly than the WL extract. HaCaT cells were pretreated with 100 mg/mL of an EAL or WL extract of *C. microphysa* for 2 h; then, they were treated with 100 ng/mL LPS (L) for 24 h. The collected cellular media were applied to (**A**) IL-1β, (**B**) IL-6, (**C**) IL-8, and (**D**) TNF-α ELISA assays. (* $p < 0.05$ vs. L group, # $p < 0.05$ vs. L + EAL group).

In addition to the anti-inflammatory effects of the *C. microphysa* EA extract in keratinocytes, we also designed a collagen homeostasis experiment to detect the physiological activity of the EAL extract of *C. microphysa* on the human fibroblasts, WS1. In the extracellular matrix, collagen homeostasis mainly depends on the balance between collagen production and breakdown. Therefore, we investigated the effects of the *C. microphysa* EA extract on the production of type I collagen and the secretion of type I procollagen and MMP-1 from WS1 cells, respectively. WS1 cells were treated with a 0–100 mg/mL EA extract of *C. microphysa* for 24 h; then, we collected the cellular medium. The data of the ELISA indicated that the *C. microphysa* EA extract dose-dependently elevated the type I procollagen secretion but decreased the MMP-1 level in the WS1 cell medium (Figure 4). Furthermore, according to the result of the Western blot, the *C. microphysa* EA extract dose-dependently triggered type I collagen expression up to 3.8-fold in WS1 cells (Figure 5). The above results show that the non-water-soluble compounds of *C. microphysa* contribute to collagen homeostasis in the extracellular matrix. They may have great potential for dermatological applications.

In order to further understand which potential secondary metabolites contained in *C. microphysa* EA extract may contribute to the abovementioned anti-inflammation and collagen homeostasis, we applied UHPLC-QTOF-MS analysis in positive ionization mode to identify the *C. microphysa* EA extract. The total ion chromatogram of the EA extract of *C. microphysa* showed several peaks, suggesting the presence of several components in the extract (Figure 6A). The data were processed with Agilent MassHunter Qualitative Analysis Software (Version B.07.00), using the Find by Formula algorithm to view the output data after instrumental analysis. After the suspect screening, four terpenoid metabolites of the *C. microphysa* EA extract were identified, including siphonaxanthin ($C_{40}H_{56}O_4$), caulerpenyne ($C_{21}H_{26}O_6$), caulerpal A ($C_{17}H_{20}O_4$), and caulerpal B ($C_{16}H_{20}O_3$) (Figure 6C–F).

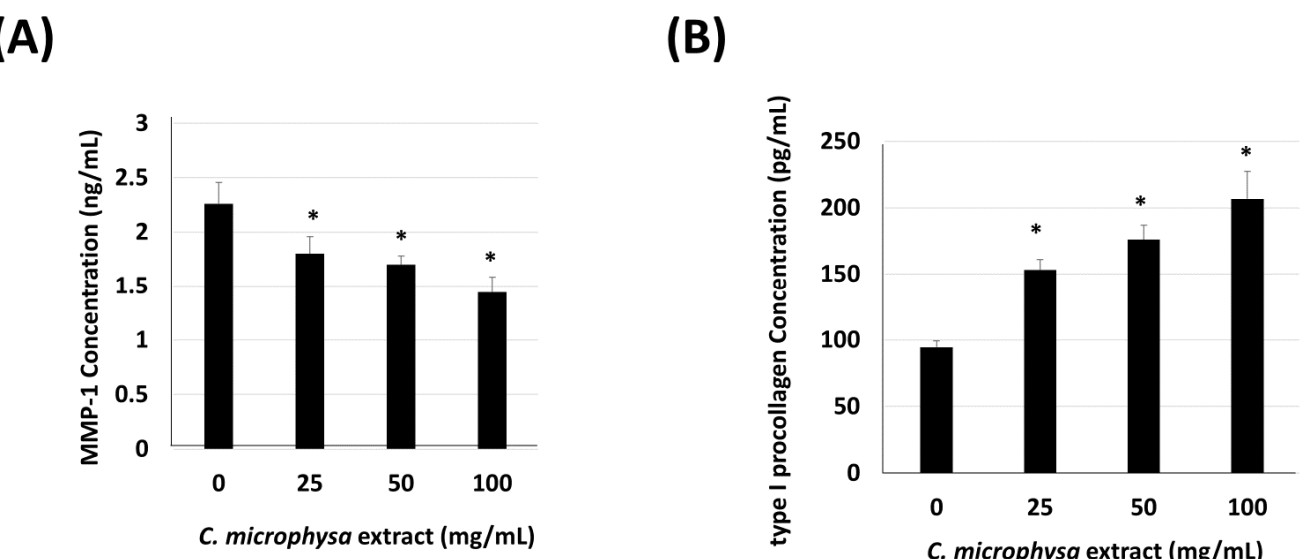

**Figure 4.** EAL extract of *C. microphysa* dose-dependently decreases MMP-1 and promotes type I procollagen secretion. WS1 cells were treated with various doses of the EAL extract of *C. microphysa* for 24 h. The collected cellular medium was applied to (**A**) MMP-1 and (**B**) type I pro-collagen specific ELISA. (* $p < 0.05$ vs. corresponding zero control).

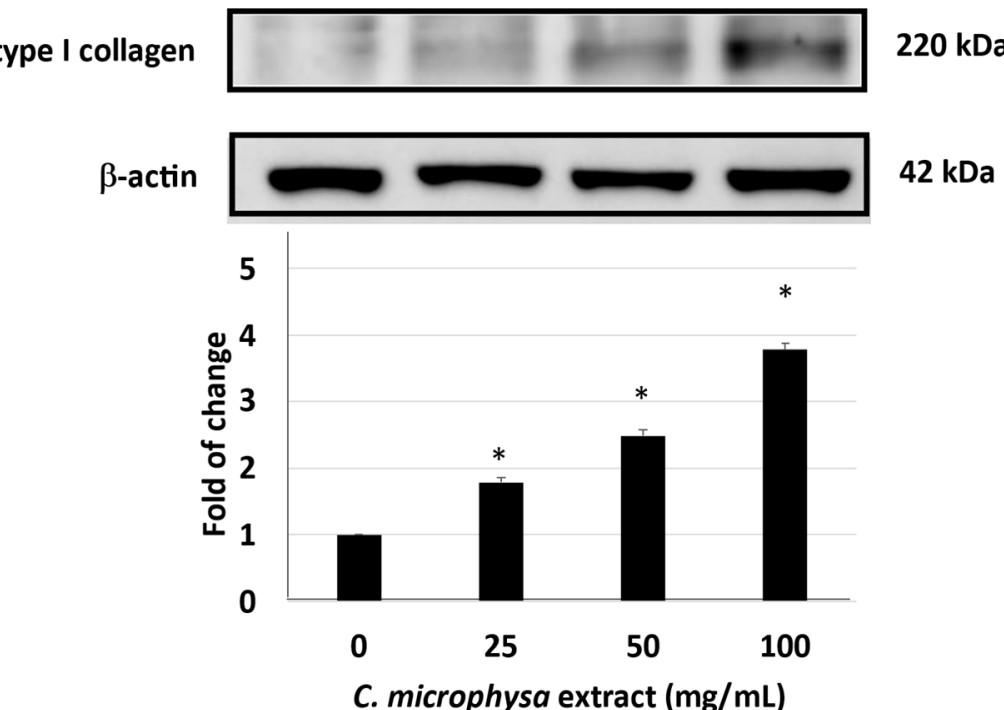

**Figure 5.** EAL extract of *C. microphysa* promotes type I collagen expression. WS1 cells were treated with various doses of the EAL extract of *C. microphysa* for 24 h. The collected cellular lysate was applied to a Western blot assay with type I collagen and β-actin antibodies. (* $p < 0.05$ vs. zero control).

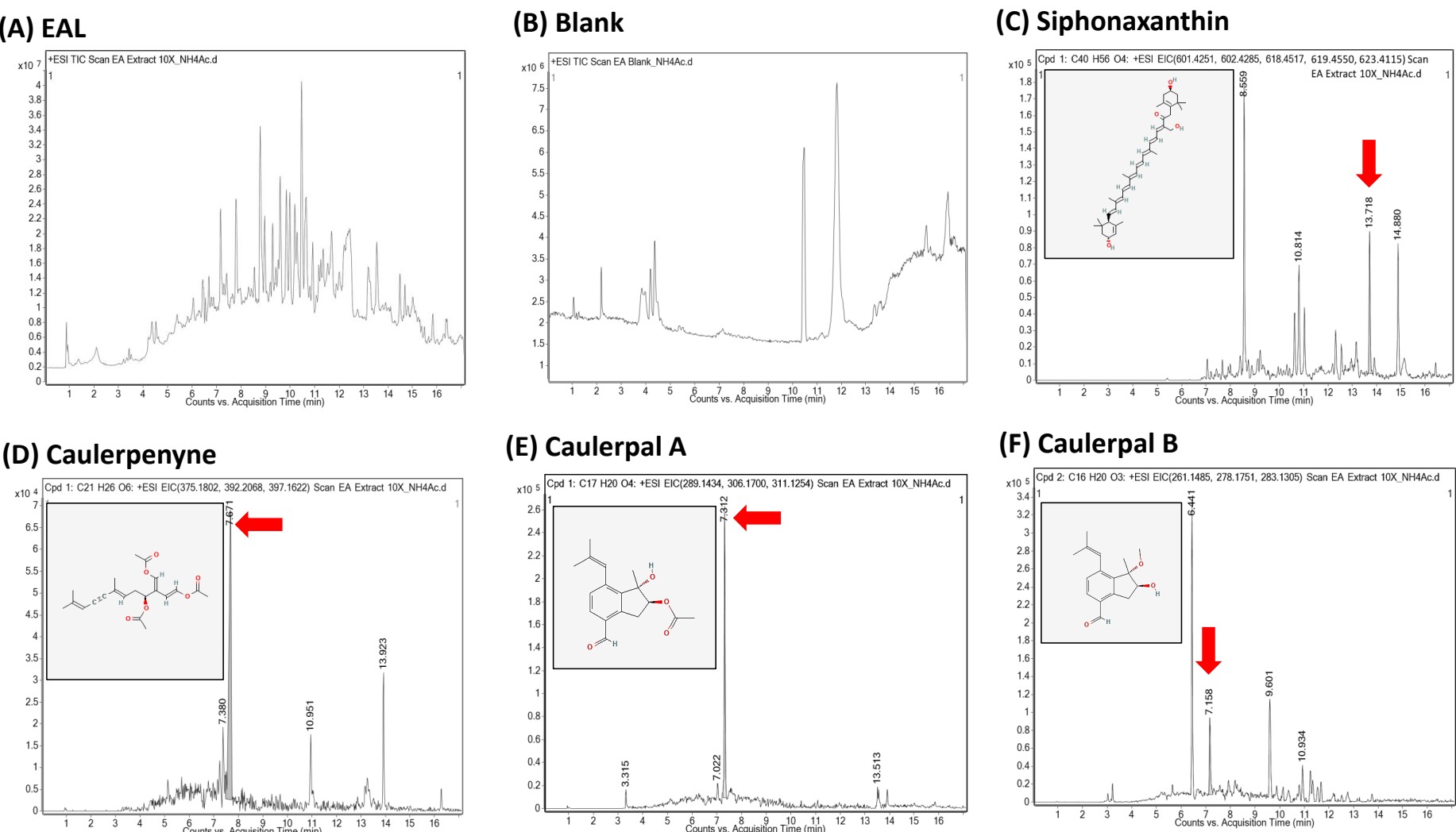

**Figure 6.** Four terpenoid compounds are present in *C. microphysa* EA extract. The UHPLC-ESI-QTOF-MS analysis in the positive electrospray ionization mode shows the chromatogram intensity against the acquisition time; (**A**) EA extract of *C. microphysa*, (**B**) blank (EA), (**C**) siphonaxanthin, (**D**) caulerpenyne, (**E**) caulerpal A, and (**F**) caulerpal B. The four marker ingredients are indicated by red arrows, and the structures are shown in the upper left.

## 4. Discussion

This is the first study that demonstrated that *C. microphysa* EA extract effectively inhibited the LPS-induced inflammatory response in human skin keratinocytes and promoted collagen homeostasis by human fibroblasts.

Collagen is the main protein of the ECM. Collagen homeostasis, the balance between synthesis and degradation, is tightly regulated for the structural integrity of the ECM and is critical for tissue physiological homeostasis and repair [44]. Type I collagen is the most abundant ECM collagen in the skin, bones, and connective tissues [29], which is degraded during various normal physiological processes, such as wound healing and skin aging, and pathological conditions such as inflammation, atherosclerotic cardiovascular disease, and sunburn [35–37]. MMP-1, also known as interstitial collagenase, is primarily responsible for the degradation of dermal type I collagen. It is secreted by unstimulated fibroblasts involved in collagen homeostasis and upregulated by inflammation and photoaging [40–42].

Inflammation is a series of protective responses for the host defense system of the skin, which is exposed to various external stimuli throughout life. Even in physiological cellular aging, the pro-inflammatory cytokine, IL-1α, was upregulated in senescent keratinocytes [45]. The chronic inflammatory condition in senescent skin accelerates collagen loss and skin thinning. Further, immune cells activated by inflammation or other stimulators produce proteases (e.g., MMPs), cytokines (e.g., IL-1β, IFN-α), growth factors (e.g., TGF-β), and ECM components (e.g., fibronectin) and then participate in the remodeling of ECM components [46,47]. On the other hand, collagen peptides were found to suppress the LPS-induced inflammation in fibroblasts and keratinocytes [47]. The above experiments seem to imply that there is dynamic crosstalk between anti-inflammation and collagen stability in skin. Most anti-inflammatory studies on green algae have focused on the activity of its polysaccharides [26,48], but there has been a lack of research on the non-water-soluble components of green algae. In this present study, we demonstrated that the water-insoluble extract of *C. microphysa* not only has superior anti-inflammatory effects in keratinocytes but also has great potential to stimulate fibroblasts to express type I collagen and to decrease the collagen-degrading enzyme, MMP-1, secretion. Therefore, regardless of the causal relationship between anti-inflammation and collagen stabilization, the EA extract of *C. microphysa* is clearly beneficial for collagen homeostasis. And the skin-protective effects of *C. microphysa* EA extract imply the pharmacological and cosmetic application potential of this edible green alga.

In addition, in order to explore the possible secondary metabolites responsible for the beneficial effects on anti-LPS-induced inflammation and collagen production in skin cells, we analyzed the *C. microphysa* EA extract by UPLC with QTof and identified four terpenoids. They are siphonaxanthin, caulerpenyne, caulerpal A, and caulerpal B, respectively. Siphonaxanthin is a specific keto-carotenoid, which was isolated from green algae and is found to suppress advanced glycation end product-induced inflammatory responses in RAW264 macrophages [19]. It also has been shown to have angiogenesis-inhibiting and lipid metabolism-regulating activities [18,20,21]. Caulerpenyne is a sesquiterpene metabolite of *Caulerpa* and was extracted from *Caulerpa taxifolia,* which exhibits antiproliferative effects on the SK-N-SH tumor cell line and alters the structure of the microtubule network [49]. It also shows antioxidant, antiproliferative, antibiotic, and antiviral activities [13,22–24]. Caulerpal A and B are the sesquiterpenes with a unique aromatic valerenane-type carbon structure, and they exhibit a broad spectrum of antifungal activity [25]. However, standards for these four terpenoids are not readily available; so, we have no further insight into their content in *C. microphysa* EA extracts. Furthermore, the contributions of these four terpenoid compounds in the anti-inflammatory and collagen-homeostasis effects of *C. microphysa* EA extract remain to be further clarified.

For a long time, research on the physiological activities of green algae has mostly focused on water-soluble components. However, compared to the water-soluble extract, the non-water-soluble extract not only avoids the interference of salt on osmolarity but usually contains a variety of antioxidants, pigments, and fragrance components. Through

this study, we also proved the excellent anti-inflammatory and anti-wrinkle effects of *C. microphysa* EA extract. Therefore, for the cosmetics industry, the non-water-soluble extract of *C. microphysa* is a potential raw material with multiple functions.

In summary, the present study was the first to verify the superior anti-inflammatory activities in keratinocytes and type I collagen-homeostasis enhancing effects in fibroblasts of the water-insoluble extract of *C. microphysa,* which contains several interpenoids, including siphonaxanthin, caulerpenyne, caulerpal A, and caulerpal B. This is the first time the potential of the non-water-soluble extract of *C. microphysa* for dermatological and cosmetic applications has been suggested.

**Author Contributions:** Conceptualization, K.-Y.L. and H.-H.H.; Validation, Z.-C.H.; Investigation, Z.-C.H. and Z.-Y.C.; Writing—original draft, K.-Y.L. and H.-H.H.; Writing—review and editing, L.-C.C. and C.-W.W. All authors have read and agreed to the published version of the manuscript.

**Funding:** This research received a grant from the College of Medicine, National Taiwan University (111M116).

**Data Availability Statement:** The data presented in this study are available on request from the corresponding author.

**Acknowledgments:** In addition, thanks to Thalassa Mystery Biotech Co., Ltd. for providing *C. microphysa* for this study.

**Conflicts of Interest:** The authors declare no conflicts of interest.

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
