# Peer review of "The Ethyl Acetate Extract of Caulerpa microphysa Promotes Collagen Homeostasis and Inhibits Inflammation in the Skin"

_cimb, doi:10.3390/cimb46030170_

Round 1

Reviewer 1 Report (Previous Reviewer 1)

Comments and Suggestions for Authors

The authors have responded to all my previous comments. however, one comment still needs to be answered before acceptance. The molecular weight of the protein in the western blot is missing. I will suggest adding a specific molecular weight of protein.

Author Response

The authors have responded to all my previous comments. however, one comment still needs to be answered before acceptance. The molecular weight of the protein in the western blot is missing. I will suggest adding a specific molecular weight of protein.

Thanks for your crucial suggestion. We have added the molecular weight in Fig. 5.

Reviewer 2 Report (New Reviewer)

Comments and Suggestions for Authors

Dear Autors

The manuscript describes biological activities of ethyl acetate extract of Caulerpa microphysa (C. microphysa):  anti-inflammatory effects by inhibiting lipo- 27 polysaccharide-induced interleukin (IL)-1, IL-6, IL-8 and tumor necrosis factor-α secretion. The authors did a lot of work and obtained interesting results. Unfortunately, the quality of the content is very poor. The reader quickly loses orientation in the content. There is a lot to improve.

The abstract contains the Latin name of the plant and the English name in the introduction, please enter both the English and the Latin name in both parts. The family name is missing.

lines 28-29-impossible to understand

Line 29- The sentence does not start with And

line 46- after etc. should be citation, what othe compounds are detected

there are many grammatical errors in the text - I suggest language correction, like:  line 47 - research show 

59 - Culerpa is a name of a plant (italics) or compound- it is no clear

line - Caulerpa Taxifolia??????????????? realy????? capital leters in both words??? 

2.2. Preparation of Caulerpa microphysa extracts- how the species was identified

Figure 4 - y-axis - missing space before the bracket

Please write "in vivo" and "in vitro" in italics in the whole manuscript

Please change mg/ml to mg/mL in the whole manuscript, especialy in figures 1-4

Author Contributions:...... is missing

Funding:..... is missing

Conflicts of Interest:..... is missing 

Author Response

Reviewer 2

The manuscript describes biological activities of ethyl acetate extract of Caulerpa microphysa (C. microphysa):  anti-inflammatory effects by inhibiting lipo- 27 polysaccharide-induced interleukin (IL)-1, IL-6, IL-8 and tumor necrosis factor-α secretion. The authors did a lot of work and obtained interesting results. Unfortunately, the quality of the content is very poor. The reader quickly loses orientation in the content. There is a lot to improve.

The abstract contains the Latin name of the plant and the English name in the introduction, please enter both the English and the Latin name in both parts. The family name is missing.

Thanks for your crucial comment. We have rewritten the article as lines 24 and 64 in the revised version.

lines 28-29-impossible to understand

Thanks for your crucial comment. We have rewritten the article as lines 26-32 in the revised version.

Line 29- The sentence does not start with And

Thanks for your crucial suggestion. We have rewritten the article as lines 26-32 in the revised version.

line 46- after etc. should be citation, what othe compounds are detected

Thanks for your crucial suggestion. We have rewritten the article as lines 44-46 in the revised version.

there are many grammatical errors in the text - I suggest language correction, like:  line 47 - research show

Thanks for your crucial suggestion. We have rewritten the article as line 47 in the revised version.

59 - Culerpa is a name of a plant (italics) or compound- it is no clear

Thanks for your crucial reminder. We have rewritten the article as line 59 in the revised version.

line - Caulerpa Taxifolia??????????????? realy????? capital leters in both words???

Thanks for your crucial reminder. We have rewritten the article as line 61 in the revised version.

2.2. Preparation of Caulerpa microphysa extracts- how the species was identified

Thanks for your crucial question. We have added the sentence at lines 118-119 in the revised version.

Figure 4 - y-axis - missing space before the bracket

Thanks for your crucial suggestion. We have added space in the y-axis in Fig. 4.

Please write "in vivo" and "in vitro" in italics in the whole manuscript

Thanks for your crucial suggestion. We have rewritten "in vivo", "in vitro" and “ex vivo” in italics in the whole manuscript.

Please change mg/ml to mg/mL in the whole manuscript, especialy in figures 1-4

Thanks for your crucial suggestion. We have changed ml to mL in the whole manuscript and Fig. 1-4.

Author Contributions:...... is missing

Thanks for your crucial reminder. We have added the statement at lines 329-331 in the revised version.

Funding:..... is missing

Thanks for your crucial reminder. We have added the statement at line 333 in the revised version.

Conflicts of Interest:..... is missing

Thanks for your crucial reminder. We have added the statement at line 337 in the revised version.

This manuscript is a resubmission of an earlier submission. The following is a list of the peer review reports and author responses from that submission.

Round 1

Reviewer 1 Report

Comments and Suggestions for Authors

1.      The abstract needs more details on methodologies for cellular viability, cytokines, and collagen markers, compromising study reproducibility.

2.     The abstract asserts the superior anti-inflammatory effect of Caulerpa microphysa EA extract without robust comparative analysis, weakening its significance.

3.      Quantitative data supporting EA extract's superior effects are missing, impacting the study's scientific rigor.

4.      The abstract lacks detailed information on the structures, concentrations, and interactions of identified secondary metabolites, hindering a comprehensive understanding of their biological activities.

5.      Broad claims about potential applications lack depth without exploring study limitations and context.

6.      The introduction briefly mentions algae metabolites but lacks a comprehensive review, weakening the research foundation and novelty.

7.      Introduction lacks details on the extraction comparison, impacting assessment and reproducibility.

8.      Introduction lacks specific parameters for anti-inflammatory and collagen-maintaining assessments, hindering reader understanding.

9.      Claims about EA extract applications lack robust experimental support and nuanced discussion of limitations.

10.   Cell culture information is insufficient; more details on cell passage, culture conditions, and line variability issues are needed.

11.   Extraction process details are brief; more information on rationale and potential impacts on extracts would enhance clarity.

12.   Extraction process lacks yield information, essential for assessing extraction efficiency.

13.   UPLC-QToF conditions need more detail for reproducibility, including mass range, resolution, and calibration standards.

14.   Ensure consistency in units, as different sections use varying units for concentration.

15.   WST-1 analysis lacks experimental setup details; including controls and statistical analysis would improve reliability.

16.   ELISA and Western blot descriptions are brief; more information on kits, detection limits, and modifications is needed.

17.   Discussion lacks a robust comparative analysis, referencing other studies or standards for context and validation.

18.   Discussion omits addressing study limitations, such as extraction method variations and cell line-specific responses.

19.   Results on identified secondary metabolites lack thorough analysis; discuss their mechanisms and interactions for a comprehensive understanding.

20.   The discussion lacks a section on future directions, which would add value by suggesting potential avenues for further research.

Reviewer 2 Report

Comments and Suggestions for Authors

Journal Name:            CIMB (ISSN 1467-3045)

Special Issue: Natural Products and Their Biological Activities

Manuscript ID: cimb-2780375

Type: Communication

Title: The ethyl acetate extract of Caulerpa microphysa promotes collagen homeostasis and inhibits inflammation in the skin

REVIEWER’S COMMENT

Major revision comments

The manuscript is well prepared and presented. It is focused on biological properties of water-insoluble secondary metabolites of the green algae Caulerpa microphysa. Biomarkers related to cell viability, inflammation, and collagen metabolism were evaluated and tested on cell culture models. The following omissions are noted.

Abstract and Introduction

The aim of the study was not clearly stated and needs further editing. In the abstract was mentioned that “… cellular viability and proinflammatory cytokines levels were evaluated by WST-1 and enzyme-linked immunosorbent assay (ELISA) respectively. Collagen-associated markers….” (Line 15-18), without specifying in more details, what kind proinflammatory and collagen-related markers are tested.

In the Introduction (Line 54-58) the purpose is еxtended with comparison studies. It is not clear what is meant by the phrase collagen maintaining effect. The aim of the study must be clearly and specifically stated.

At the end of the Introduction section the novelty of the study should be marked.

Material and Methods

1.                       At what temperature was concentrated the EA extract?

2.                       What type collagen biomarkers were evaluated with type-1 collagen kits?

3.   How was evaluated the cell viability? This should be explained and added to M&M section. What concentrations LPS were applied?

4.   The ions that have been tracked for the qualitative analysis should also be presented. Has quantitative analysis of the extract been conducted? If it has been carried out it should be explained.

5.   Is the chromatographic method used for metabolite analysis a development of the researchers or a modification of an already published method. If it is a development of the team, data on the validation of the method should be provided, if it is a modification of a previously published method, the literature source should be duly cited.

Results

1.     Figure 1. “acetate” layer should be replaced with “ethyl acetate” layer.

2.     Please explain why you consider that “…water-insoluble secondary metabolites have great potential for anti-aging and skin repair applications”? You have not tested the metabolites itself, but the whole EA extract.

3.     Figure 6. Chromatograms of a blank sample are also recommended to be included for greater certainty. The chemical structures and the Mm of these metabolites should also be given.

4.     From the chromatogram in Fig. 6 is visible that peaks corresponding to S1, S3, and S4 are not well resolved. How would you comment the peak purity? The evaluation of the matrix effect should be explained and discussed.

Discussion

1.     The conclusion is not consistent with the results presented. It is not appropriate to draw conclusions about the effect of these water-insoluble metabolites, since their effects on the inflammatory process and on the metabolism of type 1 collagen have not been studied either individually or in combination. In addition to these 4 metabolites, the extracts contain a number of other biologically active substances that also contribute to the overall biological effect.

2.     It is advisable to highlight the novelty in this study as well as its limitations and strengths.

Overall Recommendation

The paper can in principle be accepted after major revision and editing.

Comments on the Quality of English Language

Major editing required.